# Optimization of Lipid Nanoparticles by Response Surface Methodology to Improve the Ocular Delivery of Diosmin: Characterization and In-Vitro Anti-Inflammatory Assessment

**DOI:** 10.3390/pharmaceutics14091961

**Published:** 2022-09-16

**Authors:** Elide Zingale, Salvatore Rizzo, Angela Bonaccorso, Valeria Consoli, Luca Vanella, Teresa Musumeci, Angelo Spadaro, Rosario Pignatello

**Affiliations:** 1Department of Drug and Health Sciences, University of Catania, 95125 Catania, Italy; 2NANOMED—Research Centre for Nanomedicine and Pharmaceutical Nanotechnology, University of Catania, 95125 Catania, Italy; 3CERNUT—Research Centre for Nutraceuticals and Health Products, University of Catania, 95125 Catania, Italy

**Keywords:** flavonoids, poor water solubility, experimental design, degenerative eye disease, ARPE-19 cells

## Abstract

Diosmin is a flavonoid with a great variety of biological activities including antioxidant and anti-inflammatory ones. Its cytoprotective effect in retinal pigment epithelium cells under high glucose conditions makes it a potential support in the treatment of diabetic retinopathy. Despite its benefits, poor solubility in water reduces its potential for therapeutic use, making it the biggest biopharmaceutical challenge. The design of diosmin-loaded nanocarriers for topical ophthalmic application represents a novelty that has not been yet explored. For this purpose, the response surface methodology (RSM) was used to optimize nanostructured lipid carriers (NLCs), compatible for ocular administration, to encapsulate diosmin and improve its physicochemical issues. NLCs were prepared by a simple and scalable technique: a melt emulsification method followed by ultrasonication. The experimental design was composed of four independent variables (solid lipid concentration, liquid lipid concentration, surfactant concentration and type of solid lipid). The effect of the factors was assessed on NLC size and PDI (responses) by analysis of variance (ANOVA). The optimized formulation was selected according to the desirability function (0.993). Diosmin at two different concentrations (80 and 160 µM) was encapsulated into NLCs. Drug-loaded nanocarriers (D-NLCs) were subjected to a physicochemical and technological investigation revealing a mean particle size of 83.58 ± 0.77 nm and 82.21 ± 1.12 nm, respectively for the D-NLC formulation prepared with diosmin at the concentration of 80 µM or 160 µM, and a net negative surface charge (−18.5 ± 0.60 and −18.0 ± 1.18, respectively for the two batches). The formulations were analyzed in terms of pH (6.5), viscosity, and adjusted for osmolarity, making them more compatible with the ocular environment. Subsequently, stability studies were carried out to assess D-NLC behavior under different storage conditions up to 60 days, indicating a good stability of NLC samples at room temperature. In-vitro studies on ARPE-19 cells confirmed the cytocompatibility of NLCs with retinal epithelium. The effect of D-NLCs was also evaluated in-vitro on a model of retinal inflammation, demonstrating the cytoprotective effect of D-NLCs at various concentrations. RSM was found to be a reliable model to optimize NLCs for diosmin encapsulation.

## 1. Introduction

Inflammation and oxidative stress play an important role in the pathogenesis of many retinal degenerative diseases, such as diabetic retinopathy (DR) and age-related macular degeneration (AMD). In such diseases, the control of the main homeostatic mechanisms that in normal conditions counteract the relevant production of reactive oxygen species (ROS), the consequence of the continuous exposure to light stimuli, and visual signal transduction pathways are lost and stress is uncontrolled [1,2,3,4]. The lack of control leads to an imbalance between pro- and antioxidative signalling and leads to disproportionate oxidative stress, inflammation, dysregulated immune responses, and in some cases a potential impairment of the blood–retinal barrier (BBB) and tissue damage [1,5].

The treatment of eye diseases, especially those affecting the posterior segment of the eye, is still complicated nowadays. Drug diffusion to the deeper tissues is not always guaranteed and invasive treatments are often required. The use of topical treatments, now increasingly studied, is problematic, especially since many of the molecules to be used are poorly soluble and bioavailable. Nanotechnology tools provide several key benefits, including the ability to overcome eye barriers, improve drug residence time on the corneal surface, increase the permeability and bioavailability of drugs, reduce the degradation of unstable drugs, and be well tolerated by the patient [6,7].

Polyphenols, flavonoids, and natural compounds with antioxidant effects have been shown to have benefits for visual function. These molecules may in fact be able, through specific pathways, to slow down and prevent the progression of ocular degenerative diseases. The most interesting flavonoid effects in such diseases are exerted on inflammation and oxidative stress pathways, where they are able to suppress the harmful effect of ROS. Involved in these pathways are proteins such as Sirt-1 that play a key role in the processes of ageing, apoptosis, inflammation, and cellular protection from oxidative stress-induced damage. Retinal degenerative diseases have been associated with downregulation of Sirt-1 [8,9]. For this reason, antioxidant molecules and flavonoids with an agonist role for SIRT-1 activity are gaining importance as a promising therapeutic strategy in the treatment/prevention of chronic eye diseases [10,11,12,13].

Diosmin (diosmetin 7-O-rutinoside) is a naturally occurring flavone glycoside that has proven useful in the treatment of inflammatory diseases. It is obtained by dehydrogenation of hesperidin, and is abundant in the pericarp of several citrus fruits, but it can also be found in *Barosma betulina* and *Ruta graveolens*, L., both belonging to the Rutaceae family [14]. It has been shown to improve vascular resistance to inflammatory processes, thus the reason behind its use for the treatment of several circulatory system illnesses [15]. Moreover, due to its bioflavonoid nature, it exerts potent antioxidant, antihyperglycemic, and anti-inflammatory activity [16,17]. Despite its potential benefits, the molecule is very difficult to deliver due to its poor solubility and poor membrane permeability: it is classified as a BCS-class IV compound [18].

In order to solve the numerous issues from the pharmaceutical perspective and enhance diosmin bioavailability, the development of nanoparticles seems to be an attractive new strategy [19,20]. A recent study reports the benefit of solid lipid nanoparticles (SLN) to treat hepatocellular carcinoma, whereby the bioavailability of diosmin is improved, revealing a strong antioxidant activity [21].

To the best of our knowledge, no study to date reports nanotechnological formulations with diosmin for ocular delivery. Therefore, our aim is to propose a useful platform for a topical ocular delivery of diosmin. Blank nanostructured lipid carriers (NLCs) were prepared in this study by a hot-melt emulsification method followed by ultrasonication. Once the NLCs were optimized using a quality by design (QbD) approach, they were loaded with two different concentrations of diosmin (D-NLCs). The optimized formulation was subjected to a deep physicochemical investigation in terms of particle size, polydispersity index (PDI), pH, viscosity, and osmolarity. Stability studies of up to 60 days at different storage conditions were carried out. Finally, the formulations were tested on a human retinal pigment epithelial cell line (ARPE-19) to assess their compatibility with as a model of retinal tissue.

## 2. Materials and Methods

For the preparation of lipid nanoparticles, Gelucire^®^ 44/14 and Capryol^®^ 90 were gifted by Gattefossé Italia s.r.l. (Milan, Italy). Softisan^®^ 100 was gifted by IOI Oleo GmbH (Oleochemicals, Hamburg, Germany). Tween^®^ 80 (Polysorbate 80) and solvents were purchased from Merck GmbH (Darmstadt, Germany). Micronized diosmin (90% purity) was used as the active ingredient (Farmaceutici Procemsa, Nichelino, Italy).

### 2.1. Experimental Design

Box-Behnken Design (BBD) was employed for NLC optimization (Design Expert^®^ 13.0, Stat-Ease Inc., Minneapolis, MN, USA). Categorical and numerical factors of design were considered to optimize the size and PDI of the blank NLC (responses). The independent variables were the concentration (% w/v) of the solid lipid (Softisan^®^ 100 or Gelucire^®^ 44/14), of the liquid lipid (Capryol^®^ 90), and of the surfactant (Tween^®^ 80) as numeric variables and the types of solid lipid (Gelucire^®^ 44/14 and Softisan^®^ 100) as categoric variables. The design consisted of three levels (high, medium, low) for the numeric variables and two levels for categoric variables and predicted 26 runs. The factors (X_1_-X_4_) and levels used in the experiments are listed in Table 1.

Contour plots and three-dimensional (3D) response surface graphs were generated to graphically represent the response values. Statistical analysis of data was performed by ANOVA, provided in the software.

### 2.2. Preparation of the NLCs

Blank NLCs were prepared by a hot-melt emulsification method followed by ultrasonication. In order to form the oily phase, the solid lipid and liquid lipid were mixed and melted at 75–80 °C under stirring. The aqueous phase, formed by water and Tween^®^ 80 at various concentrations, was brought to the same temperature as the oily phase. The aqueous phase was slowly added to the oily phase under continuous stirring (600 rpm) to produce the pre-emulsion. The pre-emulsion was sonicated, under optimized conditions, with a constant cycle for 6 min at 40% amplitude, using a probe sonicator (Branson sonifier 450, Marshall Scientific, Hampton, NY, USA). The NLC systems were cooled down for 10 min to 5 °C and then stored at room temperature. After 24 h of storage they were characterized as detailed in the next paragraph.

### 2.3. Mean Particle Size, Polydispersity Index, and Zeta Potential Measurements

The mean particle size (Z-ave) and polydispersity index (PDI) of NLC (blank and diosmin-loaded, see below) were measured using a Photon correlation spectroscopy (PCS, Zetasizer Nano S90; Malvern Instruments, Malvern, UK) at a 90° angle of detection, at 25 °C with a 4 mW He-Ne laser operating at 633 nm. Samples were prepared by diluting the NLC suspension 1:20 with distilled water. Each value was measured in triplicate. Results are shown as mean ± standard deviation (SD). The same instrument was used to measure the zeta potential (ZP) of NLCs. This value is useful as an indicator of the stability of a colloidal dispersion. The instrument made three series of up to 100 measurements to obtain an average value.

### 2.4. NLC Optimization

The optimization of the blank NLCs was performed using the desirability parameter provided by the Design-Expert^®^ (13.0.0, Stat-Ease Inc., Minneapolis, MN, USA). The desirability parameter was considered for both responses included in the experimental design: the particle size and the PDI of the NLC formulation. Desirability automatically combines all three independent variables to obtain the optimal condition of the chosen response, within the design space [22]. The desirability values range from 0 (undesirable) to one (desirable). Among the answers predicted by the experimental design, the one that had a value close to one was chosen.

### 2.5. Preparation of Diosmin-Loaded Optimized NLCs

Diosmin was loaded in the optimized formulation at two different concentrations (80 µM and 160 µM). Diosmin-loaded NLCs (D-NLCs) were prepared with the same procedure reported in Section 2.4. The drug was added to the oily phase and the preparation proceeded as described above. The formulation was placed for 10 min at 5 °C to facilitate NLC formation and subsequently stored at room temperature. Figure 1 schematically illustrates the procedure of for D-NLC preparation.

### 2.6. pH Evaluation

The pH of D-NLC formulation was determined by a Mettler Toledo pH-meter (Columbus, OH, USA). The instrument was calibrated by using buffer solutions (pH 4.01 ± 0.02; 7.00 ± 0.02 and 10.00 ± 0.02; slope 99.8%; Mettler Toledo). Prepared samples were placed in 10 mL beakers and pH was measured at 25 °C. Measurements were performed in triplicate and results are expressed as the mean ± standard deviation.

### 2.7. Osmolality Evaluation

The osmolality (mOsm/Kg) of the samples was determined using a cryoscopic osmometer (Osmomat, mod. 030-D, Gonotec, Germany). Deionized water (consistent with the 0 mOsmol point) and a 300 mOsmol/L calibration standard (consistent with the 300 mOsmol point) were used for a two-point calibration. Samples were measured in triplicate and results are expressed as the mean ± standard deviation.

### 2.8. Viscosity Analysis

The viscosity of the final D-NLC formulation was measured using a rotational viscometer, the MYR VR 3000 V1L Viscotech, (Hadamar-Steinbach, Germany). The rotor used for viscosity measurement was 1L, which is used for very low-viscosity aqueous fluids and was rotated gradually at decreasing rpm. The viscosity was then adjusted by adding dropwise hydroxypropylmethylcellulose (HPMC; 0.05%, 0.10%, or 0.15% by weight) directly in the final formulation and left to agitate for 2 h until complete solubilization.

### 2.9. HPLC Analysis

The percentage of encapsulated diosmin in the NLCs was determined after centrifugation of the samples. For this assay, 0.5 mL of each sample (D-NLCs at 80 µM and D-NLCs at 160 µM) were separated by ultracentrifuge at 10,000 rpm and 15 °C for 30 min, using 0.5-mL Ultrafree^®^-MC (Merck KGaA, Damstadt, Germany) 0.100 µm centrifugal filters (PVDF). The supernatant and pellets were analyzed separately.

HPLC analyses were performed on an Agilent 1260 Infinity II chromatographic system (Agilent Technologies, Milan, Italy) operating with a ChemStation OpenLab software (M8307AA), a quaternary pump (G7111B), a diode array detector (DAD, G7111B), and a thermostated column compartment (G1316A). Chromatographic separations were achieved using a Zorbax Eclipse XDB-Phenyl column (250 mm × 4.60 mm, 5.0 μm, Agilent). Diosmin was analyzed using an isocratic binary mobile phase consisting of (A) acetonitrile and (B) triethylamine 0.3% (v/v) in water (pH adjusted to 3.0 with formic acid acid) 25:75 (v/v). The flow rate was set at 1.0 mL/min and the column was thermostated at 24 °C during the analysis. UV spectra were recorded in the range 200–400 nm and chromatograms were acquired at 352 nm.

Stock standard solutions were prepared by weighing 10 mg of diosmin reference standard into a 100 mL volumetric flask and dissolving in a 1:1 (v/v) mixture of dimethylsulfoxide/water. Standard solutions (0.1–100 μg/mL) were prepared by dilution from the stock standard solution with the mobile phase and analyzed immediately. Eleven-point calibration curves were set up for diosmin standards to test the linearity of the UV-DAD response. Linear regression was performed, using OpenLab software, to determine slope, intercept, and the correlation coefficient (R2). The equation of the calibration curve obtained was y (µg/mL) = 0.0361x (area counts) − 0.0106 with a correlation coefficient higher than 0.99. The limit of detection (LOD, S/N = 3:1) and the limit of quantitation (LOQ, S/N = 10:1) were 0.121 and 0.054 µg/mL, respectively (n = 6).

The analytical method was validated according to ICH guidelines [23]. The acceptability criteria recommended were reached for all obtained analytical data, with overall intra- and inter-day RSD% not exceeding 6%, and accuracy values within the 95.9–104.1% range (data not shown). Identification of diosmin was determined by comparing the retention times and the UV spectra of the analyzed sample with that of a reference standard. Potential interferences from formulation matrix constituents were excluded by analyzing the chromatograms of control formulation prepared without diosmin. Furthermore, peak-purity tests, performed with OpenLab software using DAD acquired spectra, were used to demonstrate the absence of coeluting peaks [24]. Under the outlined chromatographic conditions, the retention time of diosmin was approximately 5.26 min and each run was completed within 7 min. Drug encapsulation efficiency (*EE*%) was calculated by the difference between the amount of diosmin entrapped inside the NLCs and the total amount of drug employed to prepare the nanosystems, according to the following equation:(1)$$EE\%=\frac{µg\;Diosmin\;tot.-µg\;Diosmin\;in\;the\;supernatant\;}{µg\;Diosmin\;tot}\;\times \;100$$

### 2.10. FT-IR Spectroscopy Measurements

An FT-IR spectrophotometer (Perkin-Elmer Spectrum RX I, Waltham, MA, USA) was employed for the measurement of pure diosmin, lipids (Softisan^®^ 100 and Capryol^®^ 90), and freeze-dried optimized unloaded NLCs and D-loaded NLCs. The tool was equipped with an attenuated total reflectance (ATR) accessory and a diamond window and zinc selenide crystal (diamond/ZnSe). The dried samples were mixed with dried IR-grade KBr to obtain a homogeneous mixture, which was compressed into 1 mm pellets. The background was acquired from pure KBr pellet. For Capryol^®^ 90 analysis, the liquid specimen was dissolved in dichloromethane and the solution was dripped onto a highly polished NaCl disk; the solvent was quickly evaporated, and the analysis was carried out as above. The background was acquired from pure dichloromethane. For each sample, 20 scans were collected at room temperature over the 400–4000 cm^−1^ range at a resolution of 2 cm^−1^.

### 2.11. Differential Scanning Calorimetric Analysis (DSC)

Thermal analyses were performed for neat diosmin, Softisan^®^ 100, freeze-dried NLCs and freeze-dried D-NLCs using a Mettler Toledo DSC 1 STARe system equipped with a Poly-Science temperature controller (PolyScience, Niles, IL, USA). The detection system was an HSS8 high-sensitivity sensor (120 gold–gold/palladium–palladium thermocouples) and a ceramic sensor (Mettler Full Range; FRS5) with 56 thermocouples. The calorimetric system was calibrated in terms of temperature and enthalpy changes by using indium and by following the procedure of the DSC 1 Mettler TA STARe instrument.

Each sample was accurately weighed (approx. 6 mg) and placed in an aluminum crucible and sealed using an aluminum lid by a sealing machine. The thermograms were obtained at a scanning rate of 5 °C/min over a temperature range of 20–290 °C (heating) and at a scanning rate of 10 °C/min (cooling). The thermal behaviors of the samples such as onset temperatures, melting points, and endothermic and exothermic enthalpies were extrapolated using the software provided (Mettler STARe Evaluation software system (version 13.00) installed on Optiplex3020 DELL (PolyScience, Niles, IL, USA).

### 2.12. TEM Analysis

D-NLC morphology was investigated using TEM (JEOL JEM-101). The formulation (80 mM diosmin-loaded nanoparticles) was diluted 100 times with water and then deposited on the surface of a 200 mesh Formvar^®^-coated copper grid (TAAB Laboratories Equipment, Ltd., Aldermaston, UK). The acceleration voltage was set to 190 kV. After drying, the specimens were covered with chromium prior to imaging (Quorum Q150T ES East Grinstead, West Sussex, UK).

### 2.13. NLC Stability Studies at Different Storage Conditions

Stability studies were performed complying with ICH Q1 (stability testing). All samples were evaluated after storage at different storage conditions of 4 °C, 25 °C ± 2/60% RH (relative humidity) ± 5%, and 40 °C ± 2/75% RH ± 5% (climate chamber Blinder GmbH, Tuttlingen, Germany) by measuring Z-ave, PDI, ZP, pH, and osmolarity of the samples at specific time intervals (0, 15, 30, 45 and 60 days). The experiments were performed in triplicate.

### 2.14. Cell Culture and Treatments

ARPE-19, a human retinal pigment epithelial cell line (ATCC CRL-2302, Rockville, MD, USA) was used to perform in-vitro evaluation of NLCs. Cells were cultured in Dulbecco’s modified Eagle’s medium (DMEM) supplemented with 10% FBS and 1% penicillin–streptomycin solution and maintained at 37 °C and 5% CO_2_. Cells were initially treated with diosmin dissolved in DMSO (final concentration 0.1% w/v) at different concentrations (0.1, 1, 10, 50, and 100 µM) for 48 h; subsequently, the NLC formulations were diluted in the medium (0.0025–0.5% v/v) in order to test different concentrations of empty NLCs and D-NLCs (both at 80 µM and 160 µM drug concentration); each treatment was administered and maintained for 48 h. Ultimately, cells were treated with TNF-α (20 ng/mL) to induce an inflammatory condition, then co-treatment with D-NLCs 160 µM was administered and also maintained for 48 h.

### 2.15. Cell Viability Assay

To evaluate the effect of diosmin alone and in the NLC formulations on cell viability, ARPE-19 cells were seeded in 96-well plates at a density of 10.0 × 10^3^ cells/well in 100 µL of culture medium. After 24 h cells, were treated as mentioned before for 48 h. Following treatments, 100 µL of a 0.25 mg/mL solution of 3-(4,5-dimethylthiazol-2-yl)-2,5-diphenyltetrazolium bromide (MTT) (ACROS Organics BV) in the culture medium was added to each well, and cells were incubated for 2 h at 37 °C and 5% CO_2_. After incubation, the supernatant was removed, and 100 µL of DMSO was added to each well to dissolve formazan salts produced by active mitochondria. The amount of formazan was proportionate to the number of viable cells in the sample. Finally, absorbance (OD) was measured in a microplate reader (Biotek Synergy-HT, Winooski, VT, USA) at λ = 570 nm. Eight replicate wells were used for each group. At least three separate experiments were conducted.

### 2.16. Statistical Analysis

All experiments were carried out in triplicate and results were presented as the mean ± SD. The *t*-test was used in stability studies to calculate the statistical significance. The obtained values were considered not significant at a *p*-value > 0.05 and significant with increasing influence from a *p*-value ≤ 0.05 to *p*-value ≤ 0.0001.

In the case of biological studies, the statistical significance (*p* < 0.05) of the differences between the experimental groups was determined by Fisher’s method for analysis of multiple comparisons. For comparison between treatment groups, the null hypothesis was tested by either a single-factor analysis of variance (ANOVA) for multiple groups or an unpaired *t*-test for two groups, and the data are presented as mean ± SEM.

## 3. Results and Discussion

The preparation of NLCs was performed using simple and industrially scalable methods. It was achieved in two steps, a hot emulsification followed by ultrasonication. In the first step, the lipids have to be brought to a temperature higher than their melting point. Hot emulsification enables a homogenous lipid mixture to be obtained, into which the aqueous phase containing the surfactant will then be constantly dripped.

Prior to the actual preparation, a preliminary investigation was carried out to select the independent variables and factors to be kept constant during the preparation process (solid/liquid lipid mixtures, sonication time, and power). Indeed, several parameters can influence the physicochemical properties of NLCs and must be adjusted. Four different mixtures were tested (Mygliol^®^ 812 and Softisan^®^ 100, Mygliol^®^ 812 and Gelucire^®^ 44/14, Capryol^®^ 90 and Gelucire^®^ 44/14, and Capryol^®^ 90 and Softisan^®^ 100). The highest transmittance results obtained by UV spectrophotometry (as an index of better miscibility between the lipids) were obtained for the mixtures with Capryol^®^ 90, so these three lipids (Capryol^®^ 90, Gelucire^®^ 44/14, and Softisan^®^ 100) were included in the experimental design as independent variables. A visual analysis was also conducted according to the model of Kiss and colleagues to assess the solubility of diosmin in the chosen mixtures before entering the data into the experimental design [25].

The use of sonication after the emulsification step reduces particle size, as seen in previous studies [26]. Sonication is a parameter to be carefully controlled as intense sonication can lead to a rebound effect (particle aggregation). Subsequent screening made it possible to define this through the preparation of four batches with different sonication times and power levels: independently of the time (6 or 10 min) the size and PDI did not vary, whereas using more powerful sonication, the NLCs became larger as if aggregating [27]. The speed and the amplitude (intensity of sonication) of the ultrasonication process were kept constants for all runs and have been excluded as design variables.

The type and concentration of surfactant, added in the aqueous phase, are parameters that must be taken in account, since, by decreasing the surface tension between the lipid phase and the aqueous phase, they influence the stability of the formulation. The choice of the surfactant fell on Tween^®^ 80 because, as reported in the literature, it is widely used in the formulation of ophthalmic preparations, and it is FDA-approved for ophthalmic applications [28]. In particular, the study by Leonardi et al., in which lipid carriers were prepared using different surfactants, showed that Tween^®^ 80 is particularly safe and well-tolerated compared to other surfactants [29]. Tween^®^ 80 is one of the most used non-ionic surfactants for producing lipid nanostructures due to its high hydrophilic–lipophilic balance (HLB) value (15.0) and the low concentration required for particle stabilization. With regards to surfactant concentration, the preparation was always carried out with Tween^®^ 80 concentrations in the aqueous phase up to 2% by weight, since higher concentrations could induce eye irritation.

Lipid concentrations were also chosen that together did not increase the viscosity too much, which hinders the dripping of one phase onto the other, leading to non-uniform supersaturation, slower nucleation rates, and more agglomeration of the particles and, consequently, larger particles.

For the preparation, two solid lipids were chosen for which the ability to improve the bioavailability of poorly soluble active ingredients has been demonstrated in the literature [30]. The chosen lipids, Softisan 100, Capryol 90, and Gelucire 44/14, were conceived by evaluating the literature of recent years. In particular, some recent works in ophthalmic drug delivery have been published on these materials [31,32,33]. The choice of one solid lipid over another will be a matter of choice during DoE (design of experiments). Indeed, lipids consisting of long alkyl chains (LCT) or medium alkyl chains (MCT) can be used for the preparation and, depending on the choice, the properties of the molecule vary. For example, increasing the MCT rather than LCT content has been shown to reduce the viscosity of NLCs, thereby reducing the surface tension to form smaller particles. Another example is the increase of LCT that could improve the encapsulation of MCT in a LCT matrix. Several parameters then influence the final particle characteristics, including the concentration and the ratio between the solid lipid and the liquid lipid [34].

### 3.1. Box-Behnken Design and Data Analysis

In this work, we used the DoE as a tool of the QbD approach, with the aim to design an optimized NLC formulation in terms of particles size and homogeneity. The QbD approach leads to the understanding of variables, time optimization, cost reduction, and improvement of translational research from the laboratory bench to applicable therapeutic products [35]. Once the independent variables had been chosen, the software generated 26 runs, through a random combination of the four factors inserted (X_1_, X_2_, X_3_, X_4_).

The technique of hot emulsification-ultrasonication was used for the preparation of each formulation. Entering the values corresponding to two responses, size (Y_1_) and PDI (Y_2_), revealed that the 2FI (two factors interaction) model was found to be significant. In the 2FI model, there are interactions between the factors, but no higher-order effects will occur [36,37]. The F-value of the 2FI model (18.30) and *p*-value (less than 0.001) indicate the good prediction of the model.

Table 2 shows the ANOVA of the various terms; the coefficient estimate represents the expected change in response per unit change in factor value when all remaining factors are held constant. For factors where the estimated coefficient has a negative sign, the relationship between the factor and the response is an inversely proportional relationship. Thus, for example, as the value of factors X_1_ and X_3_ increases, the value of the response Y_1_ decreases. Factors characterized by a positive sign are subject to a direct proportionality relationship with the response. Therefore, as their value increases, the value of the response will also increase. Moreover, the greater the value of the estimated coefficient, in absolute value, the greater the impact that variable has on the response.

The composition of lipid nanoparticles, such as SLN and NLC, and the process parameters are crucial points in the formulation step. Factors such as the type of lipid and its concentration must be chosen carefully, as their influence is significant on the final particle size [34,38]. Application of ANOVA made it possible to assess the influence of the independent variables on the response and the type of influence. The two polynomial equations revealing the interactions between the variables and the two responses, size (Y_1_) and PDI (Y_2_), are shown in the paragraphs below.

#### 3.1.1. Effect of Independent Variables on the Particle Size of NLCs

The high regression coefficient value (R^2^ = 0.9242) of the fitted regression model and insignificant lack of fit (*p*-value = 0.0001) indicated that the model (2FI) was highly adequate in predicting NLC particle size. In NLCs prepared with Softisan^®^ 100 and Capryol^®^ 90 as materials, the resulting relationship between the concentration of solid lipid and particle size showed an inverse proportionality, as indicated by the negative coefficient of solid lipid concentration. Thus, as the solid lipid increased, the particle size decreased.

The polynomial equations below show the influences the factors have on the response and the interactions between the factors.
(2)$$Y_{1}\;=\;+353.86\;-\;202.06\;X_{1}\;+\;281.75\;X_{2}\;-\;142.21\;X_{3}\;+\;294.43\;X_{4}\;-\;169.30\;X_{1}\;\times \;X_{2}\;+\;132.74\;X_{1}\;\times \;X_{3}\;-\;179.78\;X_{1}\;\times \;\;X_{4}\;-\;74.66\;X_{2}\;\times \;X_{3}\;+\;292.76\;X_{2}\;\times \;X_{4}\;-\;132.20\;X_{3}\;\times \;X_{4}.$$

In the equation, the synergistic effect is presented as plus sign and antagonistic effect as minus sign. The obtained polynomial equations were further supported by 3D response surfaces plots and contour plots. The figures below (Figure 2 and Figure 3) show the graphics of ANOVA to explain the influence of variables on the response. As can be seen, a wide variation in response (size) was obtained.

The particle size of the produced NLCs varied from 24.21 nm to 1960 nm. The values of the regression coefficient were higher for particle size than for PDI. The absolute value of this indicates the “degree” of the relationship (the greater the value of the coefficient, the more the variable influences the variable response). Thus, from our results the variables have a more pronounced influence on particle size. The sign of the regression coefficient b indicates the ‘direction’ of the relationship: the positive sign indicates a concordance between the variables (an increase of variable corresponds to an increase in y), while the negative sign indicates a discordance (an increase of variable corresponds to a decrease in y).

The highest coefficient value found for the solid lipid type shows a very highly significant effect. Indeed, by Figure 2, there is an evident difference between the influence of the two solid lipid on the size of NLCs. Particles obtained using Softisan^®^ 100 were within a narrow range. The variability with Gelucire^®^ 44/14 was very high. Particles made with this lipid were highly polydispersed and presented dimensions from the nanometer to the micrometer scale. We hypothesized that the variability of particle size with Gelucire^®^, and especially the production of particles with sizes up to 1 µm, was due to irregularities formed during the production process. In fact, the different composition of Gelucire^®^ could give rise to problems due to the different melting and cooling of its constituent parts (a small glyceride fraction and free PEG). This was probably due to the formation of two different phases when cooling or during sonication, or due to the presence of tensions, linked to the difference in solidification speed and formation of the nanoparticles, in turn linked to the different materials that make up Gelucire^®^ (PEG chain and glycerides) [39].

Design revealed that the solid lipid concentration, liquid lipid concentration, and solid lipid type exerted the most significant effect on mean droplet size (Y_1_). Droplet size was, however, also affected by liquid lipid concentration. As the experimental results indicated, the concentration of solid lipid and liquid lipid affected particle size. The reason for an increase in size with an increase in liquid lipid concentration could certainly be due to greater incorporation of the liquid lipid within the nanoparticles, resulting in an enlargement of the average diameter. A proper ratio of solid to liquid lipids in favor of the solid lipid allows for a greater order in the arrangement of the particle core, resulting in a decrease in size. At the same time, the inclusion of liquid lipids at the right concentrations increased the stability of adding a liquid lipid compared to a solid alone [40]. Although the impact on size was minor compared to that of the other variables, the average particle diameter decreased as surfactants increased. Tween^®^ 80 reduced the interfacial tension during the emulsification phase and, consequently, the size of the emulsified droplets [41].

#### 3.1.2. Effect of the Independent Variables on the PDI of NLC

The high regression coefficient value (R^2^ = 0.8836) of the fitted regression model and insignificant lack of fit (*p* = 0.0033) indicated that the model (2FI) was highly adequate in predicting the PDI of NLC suspensions.

The polynomial equation below shows the influence each factor has on the responses and the interactions between the factors:(3)$$Y_{2}=\;+0.5433\;+\;0.0194\;X_{1}\;-\;0.0023\;X_{2}\;+\;0.0844\;X_{3}\;+\;0.1229\;X_{4}\;+\;0.0988\;X_{1}\;\times \;X_{2}\;-\;0.0225\;X_{1}\;\times \;X_{3}\;+\;0.0741\;X_{1}\;\;\times \;X_{4}\;-\;0.0561\;X_{2}\;\times \;X_{3}\;+\;0.0163\;X_{2}\;\times \;X_{4}\;+\;0.0285\;X_{3}\;\times \;X_{4}$$

In the case of PDI, the two factors of solid lipid concentration and surfactant concentration exerted the maximum influence on the response. The PDI data suggest values between 0.311 and 1. Probe sonication, as a batch process, can lead to extreme variability in results. This also leads to a wide range of values possibly given by this variable process [42]. Once more, a large variability is given by the type of solid lipid: from the equation above, it appears that the choice of solid lipid (factor A) influences the homogeneity of the formulation. The increase of PDI and thus the loss of homogeneity of the formulation seems to be particularly associated with Gelucire^®^ 44/14. Again, the mixed composition of Gelucire^®^ (consisting of a small fraction of mono-, di-, and triglycerides, and mainly PEG-32 (MW 1500) mono- and diesters of lauric acid (C12)), could lead to variability during preparation due to different heating times, and different conformations during the emulsion phase or during the sonication phase. This would lead to different particle populations in size and shape, leading to inhomogeneity in the final formulation.

On the other hand, surfactant concentration was another variable that had great impact. ANOVA data show that increasing the surfactant concentration increases the PDI. This may be due to the fact that larger amounts of surfactant tend to give the non-uniform thickness of surfactant layers on the surface of particles, which affects the size and shape of the particles, giving inhomogeneity between them and increasing the PDI value [43,44]. The concentration of solid lipid has little influence compared to the influence of the concentration of surfactant, and a tendency to increase the PDI value if the concentration increases. This is probably due to an increase in the viscosity of the organic phase on which the aqueous phase is dripped, resulting in a slower diffusion of the organic phase into the outer phase, which may give rise to the formation of aggregates and uneven distribution of the particles [45].

Figure 4 and Figure 5 showed the interaction between factors and the influence of the independent variables on the PDI as a response.

### 3.2. NLC Optimization

In order to obtain a single formulation that possesses all the characteristics to be potentially used for ocular administration, certain requirements must be established. This can be done by means of the ANOVA study done earlier in order to understand which factors most influence responses and how these influence them. In this regard, we set several criteria for each factor considered (Table 3): maximize the solid lipid concentration; minimize the liquid lipid concentration; maintain the surfactant concentration in the previously selected range; and use Softisan^®^ 100 as a solid lipid to obtain NLCs of a size suitable for ophthalmic administration. The choice of Softisan^®^ 100 as the solid lipid was in line with the purpose since it has been used in different studies for ophthalmic formulation, as a safe and tolerated material [46]. The need to minimize size was related to the close correlation between particle size and cell interaction. Good particle passage in the intraocular tissues has been demonstrated, with particles with a size range below 200 nm, ensuring an easy passage through ocular tissues and greater retention on the ocular surface, as well as fewer irritation problems after topical application [47].

Once the criteria have been set for each parameter, the model suggests an optimized formulation in association with a given desirability. The desirability of a model represents an estimate of the deviation of the prediction from the desired value. It ranges from 0 to 1. One represents the ideal case; zero indicates that one or more answers are outside acceptable limits. From the possible formulations, the one with the highest desirability index was selected and experimentally validated. From the optimized formulations, we chose the one with the highest desirability index and at the same time not too small in size. Particles smaller than 20 nm are very likely to be drained with less retention effect on the ocular surface, due to removal by the conjunctival, episcleral, or by other periocular circulatory systems and the entry into circulation. It is claimed that particles with a size of 20 nm have a rapid decline from 77% retention dose to 15% [48]. This would lower the final bioavailability, so the choice fell on a formulation whose proposed size was at least 50 nm larger. The optimized formulation (OPT-NLC) consisted of 10% w/v Softisan^®^ 100, 3% w/v Capryol^®^ 90, 1.159% Tween^®^ 80, and a desirability value of 0.993. Figure 6 shows the contour plots for the optimized formulation predicted by the software.

As Figure 6 shows, the size predicted by the software was 80.15 nm and the PDI 0.223. DLS analysis showed that the obtained size was very close to the predicted size and PDI values, demonstrating a very good correlation of the data and a high predictivity of the model. The percentage error between the predicted size and that obtained experimentally was 6.43%; therefore, there was a good correlation between the prediction made by the model and the values obtained experimentally: an error less than 10% with respect to the predicted value in fact indicates a good process [49].

### 3.3. Technological Characterization of NLC

The optimized formulation was then characterized prior to proceeding with the encapsulation of the drug to assess its suitability for the selected administration route (Table 4).

The net negative ZP values reveal what is already known from the literature: nanoparticle systems with a lipid matrix, and NLCs in particular, are quite stable and rarely tend to aggregate. The superficial charge of lipid NPs is mostly determined by the materials used for their synthesis and by the pH of the surrounding medium. All components in our formulation were neutral or weakly acidic, such as Softisan^®^ 100 (glyceride) and Capryol^®^ 90, which has a pH value ranging from 5 to 7. Therefore, the final charge given by these components was negative but not so strong as to reach the optimum value of + or −30 mV. In any case, a not excessively high negative charge, due in part to the non-ionic nature of the surfactant, is more likely to be tolerated after instillation in the eye.

One of the key parameters for a formulation to be instilled into the eye is osmolarity. Normal homeostasis requires a regulated tear flow, the main principle of which is osmolarity. It has been suggested that tear hyperosmolarity is the main cause of discomfort, ocular surface damage, and inflammation in dry eyes [50]. The final osmolality of the NLC formulation was very low, being a formulation made by 90% of water and low in electrolytes. Such a value would lead to some eye irritation and dryness, being a hypotonic solution compared to tear fluid (about 0.300 Osm/Kg). For this reason, the osmolality of NLC was adjusted by adding decreasing concentrations of NaCl from 0.9% (w/v) downwards. With NaCl concentrations of 0.9% and 0.8% (w/v), osmolarity values of 0.379 Osm/kg and 0.320 Osm/kg were obtained, respectively. Some studies state that the tolerated osmolarity can reach 0.350 Osm/kg, but osmolarity values slightly lower than physiological are still preferable to higher ones [51]. An optimal osmolarity of 0.243 Osm/kg was therefore obtained by adding 0.7% (w/v) NaCl.

The viscosity of ophthalmic suspensions has a clear impact on ocular absorption of drugs, and is also important in assessing the tolerability of a formulation once applied. Solutions that are too viscous can cause discomfort to the eye. The viscosity of the D-NLC formulation as such was 0 mPa. It has been shown that increasing viscosity from 1 mPa up to 15 mPa resulted in an approximately fourfold increase in ocular absorption; thus, an increase in viscosity value is an important consideration when formulating ophthalmic suspensions [52]. Apparently, a higher viscosity of eye drops is able to retain suspended particles. Therefore, the analysis was repeated by adding increasing amounts of HPMC to the D-NLC formulation (0.05%, 0.10%, or 0.15% w/v). Only the latter concentration produced an optimal formulation for ophthalmic administration: D-NLC with the addition of 0.15% HPMC had a viscosity of 23 mPa at 30 rpm. Solutions with values up to 30 mPa at 30 rpm are known to be well tolerated by the human eye [53].

Figure 7 shows the pseudoplastic behavior of D-NLCs under rotational stress. One pseudo-plastic sample had a viscosity which decreased when the shear rate increased. The pseudo-plastic behavior is typical of liquids that have a very low viscosity. It is due to the shearing action on the long chain molecules of materials such as linear polymers; in this case the addition of HPMC [54].

### 3.4. Stability Studies

Stability assessment is a key step in the development of formulations for pharmaceutical use, especially if they are colloidal. In order to observe the storage stability of the samples, the NLCs were stored at different temperatures (namely, 4 °C, 25 °C/60% RH, and 40 °C/75% RH), measuring the Z-ave, PDI, ZP, osmolality, and pH as stability parameters at different time points. The values after storage at 4 °C are not shown, because these formulations were unstable at that temperature, producing a solidification of the lipid matrix. Storage in the refrigerator cannot be considered a suitable solution either, because of a very rapid and strong physical alteration of the systems.

Samples stored at room temperature and at 40 °C in a climatic chamber were monitored by DLS measurements, osmometer, and pHmeter. Figure 8 shows almost constant values of size of the samples stored at 25 °C. The stability offered by lipid particles probably derives from the fact that Softisan 100 has a distinctly non-polar characteristic and a not very high melting point, which also allows the formation of more amorphous structures and, thus, favors storage stability. The size decreased more and more as if there were a rearrangement of the lipids that changed the structure, making the particles smaller. Furthermore, a visual analysis after 45 days at 40 °C showed a phase separation that after gentle hand agitation was reconstituted to give the same initial particle size. After 60 days this phase separation struggled to reconstitute, giving larger particles than in the original system. The instability caused by the phase separation is not to be attributed to lipid incompatibility; rather to a probable polymorphism of Capryol^®^ 90 or Softisan^®^ 100, which should, however, be better evaluated in the future. Indeed, it has been shown that the more polymorphic a form is, the more it may undergo changes leading to macroscopic instability phenomena [55]. The enlargement of particle size due to storage at 40 °C may be related to an acceleration of a phenomenon called Oswald ripening that increases at high temperatures. In fact, a higher temperature leads to increased diffusion and reaction kinetics at the particle boundary layer interface and thus, faster growth [56,57]. The smaller the particles initially are, the more this phenomenon is favored.

The PDI value, however, remained around 0.3 throughout the whole test, demonstrating good particle homogeneity even in the presence of a phase separation. The zeta potential always remained negative; when the phase separation occurred, it became slightly less negative, probably because of an agglomeration of the particles. Measurements of the ZP over time show that the formulation certainly had better stability when stored at room temperature. For these samples, the ZP values seemed to stabilize over time, approaching the optimal value of 30 mV [58]. The change in zeta potential at 40 °C after about 30 days would suggest instead a partial instability of the system, with possible particle aggregation.

As shown in the graphs in Figure 8, a decrease of pH value after 30 days of storage was observed, more evident when the formulation was stored at 40 °C. The osmolarity also followed the same trend. The decrease in pH and osmolarity was probably due to an initial microbial growth. In fact, the increase in acidity could be the result of the activities of intrinsic microorganisms that lead to fermentation and the production of organic acids (microbial metabolites) over time [58]. The use of a preservative could improve performance and will be tested in forthcoming studies.

Overall, the stability of NLC samples at room temperature was good and may be due to the use of a technique (ultrasonication) that allows better organization of lipids and surfactants at the interface, resulting in a good arrangement of materials in the particles.

### 3.5. Encapsulation of Diosmin into Optimized NLCs

The optimized formulation of NLCs was loaded with diosmin at two different concentrations. The concentrations were chosen by analyzing several studies in the literature. Indeed, diosmin has been shown to have a cytoprotective effect against oxidative stress on retinal pigment epithelium cells (ARPE-19) exposed to high concentrations of glucose, demonstrating an antioxidant effect at concentrations of 0.10 micrograms/mL [59]. Furthermore, at a concentration of 24 micromolar the drug demonstrated an inhibitory effect on aldose reductase in the upregulation in diabetes complications [60].

It is known that most of the administered dose is lost and does not reach the back of the eye when administered topically. With the aim of reaching such target, and taking effective doses found in the literature as examples, we increased the concentration to be encapsulated to enhance the formulation in order to reach deeper tissues with optimal dosage. D-NLC formulations at 80 µM and 160 µM were therefore prepared and optimized (Table 5).

From Table 5 it can be seen that there were no significant changes between NLCs and D-NLCs loaded at the two different drug concentrations. This absence of variations in the physicochemical parameters and the high entrapment efficiency may support the following hypothesis: the values of EE% reflect the solubility and affinity of diosmin for the used lipids; also, the lipophilicity of diosmin with the higher amount of lipids may play a role in increasing the entrapment efficiency. A hydrophobic interaction between diosmin and the hydrocarbon chain of the esterified fatty acids in the lipids can thus be a possibility. The solubility of diosmin in these lipids and incorporation into NLCs is attributed to interactions such as hydrogen bonding or van der Waals forces or a combination of both [60]. Softisan^®^ 100 is able to solubilize the diosmin, which can be attributed to the presence of long chain fatty acids (C10–C18) [61]. This result is important in order to increase the bioavailability of diosmin. Indeed, diosmin has an ionization constant (pKa) of 10, which makes it hard to permeate across the cell membrane without an appropriate delivery means [62]. By incorporation into lipid nanoparticles, its solubility and permeation through ocular tissues can be strongly increased.

A microscopic evidence of NLC formation was achieved by TEM analysis. Figure 9 shows the presence or regular rounded structures, with a net differentiation between the inner liquid core and the external wall made by the solid lipid.

### 3.6. FT-IR Analysis and DSC Analysis Proofs of Diosmin Encapsulation

An analysis by IR spectroscopy was carried out to confirm the effective encapsulation of diosmin within the lipid nanocarrier. Figure 10 shows the spectra of lipids (Capryol^®^ 90 and Softisan^®^ 100), neat diosmin, NLCs, and D-NLCs and a physical mixture obtained by destroying D-NLCs with methanol for 1 h under magnetic stirring. From the IR spectrum of diosmin, the characteristic peaks found in the literature are evident [63,64]. In particular, the absorption bands at 3435 (cm^−^^1^) due to the 𝛼-hydroxycarbonyl group of flavonoids, around 1660 cm^−^^1^ due to aromatic ketonic carbonyl stretching (C=O vibration), and a band around 1600 cm^−^^1^ due to the C=C bond in the aromatic ring are evident. Such signals were no longer visible in D-NLCs: absence of the peaks proper of diosmin in the second case evidenced the complete encapsulation of the drug molecules within the lipid matrix. Further evidence of this was obtained when the D-NLC system was destroyed by methanol and re-analyzed: at this point the diosmin peaks were visible. In the spectra of Softisan^®^ 100, NLCs, D-NLCs, and destroyed D-NLCs the typical Softisan^®^ 100 peaks were noted, with characteristic absorption bands related to C-H stretching of benzene (2800–2955 cm^−^^1^), carbonyl compound CHO (1728, 1739 cm^−^^1^), C-O stretching (1000–1300 cm^−^^1^) [31]. Additionally, Capryol^®^ 90 typical peaks were identified at 3432 (O-H), 2925–2859 (aliphatic C-H), and 1728 (C=O) cm^−^^1^.

The IR spectrum of empty NLCs showed a substantial overlap of the spectra of the lipid constituents, with no significant shift in the main peaks [62,65]. This indicates that the formulation process did not induce any interaction or alteration of the compounds used. The spectra of blank NLCs and D-NLCs are also almost superimposable.

The DSC thermograms of raw materials and optimized NLCs and D-NLCs are given in Figure 11. The thermogram of diosmin was typical of a non-crystalline compound with a sharp endothermic peak at the onset temperature of 274.39 °C and a broad endothermic peak at 129.46 °C, corresponding to loss of water [63]. Softisan^®^ 100 exhibited a sharp endothermic signal with an onset melting point at 37.29 °C and another at 41.87 °C (Figure 11B) [61]. The disappearance of the characteristic endothermic diosmin peak in the thermograms of D-NLCs (Figure 11D) demonstrates that the drug was successfully entrapped in an amorphous state within the lipid matrix. It possible that there is a molecular distribution of diosmin in the lipid, with no aggregation of drug molecules, which means that, upon heating, diosmin molecules will appear as dissolved molecules in the melted lipid. DSC peaks of the lipids in blank NLCs and D-NLCs were shifted to a lower temperature, which may be due to lipid polymorphism and/or interaction with diosmin. Therefore, it can be concluded that NLC structures were formed, since curves C and D in Figure 11 differ from the solid lipid of origin, showing lower and less enthalpically visible transition temperatures than the Softisan^®^ 100 specimen (Figure 11A).

### 3.7. In-Vitro Biological Evaluation

In-vitro studies have been performed on retinal pigment epithelial cells (RPECs), which are important as a signal for assessing normal retinal function. RPECs are the main components of the cellular layer between choriocapillaris and photoreceptor outer segments. Their main functions include maintenance of BBB, and secretion of cytokines and chemokines, together with the release of growth and neurotrophic factors, and they contribute to the central retina’s immune defence. Due to their key role in maintaining vital eye functions, RPECs have a significant weight in the occurrence of different retinal inflammatory states [66,67]. Retinal inflammation is involved in most retinal disorders, such as diabetic retinopathy, uveitis, and age-related macular degeneration (AMD) [68].

Tumour necrosis factor-α (TNF-α) has a pivotal role in ocular inflammation and its interaction with the retinal pigment epithelium (RPE) seems to be related to all of the ocular abovementioned disorders [69]. TNF- α promotes inflammation impacts directly on cellular homeostasis inducing cytotoxicity and enhancing the release of pro-inflammatory cytokines, arachidonic acid intermediates, ROS, and angiogenic factors [70]. Furthermore, TNF-α negatively regulates the transcription of factor orthodenticle homeo-box 2 (OTX2), which is often expressed in the RPE and maintains the outer retina’s homeostasis balance [71]. Since the main aim of this study was to design a formulation suitable to treat diseases of the posterior eye segment, ARPE-19, a spontaneously arising RPE cell line derived from the normal eyes of a 19-year-old male, were chosen as a model for in-vitro studies of both cytocompatibility and cell viability following TNF-α -induced damage.

As shown in Figure 12, diosmin treatment reduced cell viability in a dose-dependent manner; in particular, the highest tested concentration (100 µM) resulted in 54% cell death. In order to evaluate the eventual cytotoxicity of empty NLCs and drug loaded formulations, cells were treated with different concentrations of culture medium-diluted NLCs/D-NLCs (Figure 13). Interestingly, NLCs showed no significant reduction of cell viability. This result was attributable to the fact that the materials used for NLCs (Softisan^®^ 100, Capryol^®^ 90, and Tween^®^ 80) are very biocompatible with ocular tissue. They have already been tested in ophthalmic formulation delivery work and found to be non-toxic and well-tolerated [72,73]. However, D-NLCs 80 µM caused a slight decrease only at high concentrations (0.25–0.5 % v/v) while D-NLCs 160 µM was already cytotoxic at lower concentrations (0.025 % v/v) (Figure 12). The increase in cell viability and, consequently, the cytoprotective effect of diosmin, could be due to its anti-apoptotic effect, which has also been demonstrated on ARPE-19 cells [74,75].

The anti-inflammatory effect of diosmin has already been studied. It is known that it inhibits pro-inflammatory factors in cases of diseases affecting the circulation. In fact, the most commonly used field is applied in the pharmacotherapy of chronic venous disorders [76]. To investigate the cytoprotective effect of D-NLCs in an in-vitro model of retinal inflammation, ARPE-19 cells were exposed to TNF-α 20 ng/mL and D-NLCs 160 µM for 48 h. Cell exposure to TNF-α stress is aimed at the fact that this formulation is developed to reach the retina and thus promote the anti-inflammatory effect of diosmin. TNF-α is a major cytokine involved in retinal inflammation and its expression plays a significant role in the development of retinopathies. Therefore, its inhibition and regulation are an important target in alleviating retinal inflammatory diseases. The development of a cell viability assay following this stimulus mimics the expression of this mediator in retinal cells. By inhibiting this pro-inflammatory cytokine, diosmin could reduce biochemical and histological changes that are revealed in the retinal area, for instance, following pathological stress [77]. Furthermore, the study by Shalkami et al. demonstrated a dose-dependent reduction in TNF-α levels in a different model, treated with diosmin [17].

TNF-α caused a reduction of cell viability (41%) (Figure 14) which was slightly but significantly reversed by D-NLCs 160 µM at the non-cytotoxic concentrations 0.005–0.075–0.01 % v/v.

## 4. Conclusions

The aim of this work was to design a nanotechnological formulation compatible with the ocular administration of diosmin and its targeting to the posterior part of the eye. A Box-Behnken design was found to be a reliable model to optimize NLCs with suitable properties. The optimized formulation was subjected to a physicochemical and technological investigation revealing that NLCs present an average size of about 85 nm and a net negative surface charge (−18.5 mV). The formulation was adjusted in terms of pH and osmolarity, making it highly compatible with the ocular tissues. The optimized formulation was encapsulated with diosmin (D-NLCs) at two drug concentrations (80 and 160 µM), reaching an encapsulation efficiency of 99% in both cases. The formulations were shown to be very stable at room temperature for at least 6 months. D-NLCs were found to be safe and well-tolerated by ARPE-19 cells and could be a promising candidate for the treatment of ocular tissues inflammation.

With this aim, additional studies are ongoing to further assess and confirm D-NLCs’ anti-inflammatory activity. Moreover, since another aspect of ocular degenerative diseases is the overproduction of reactive oxygen species (ROS), further studies will be done to evaluate the antioxidant activity of loaded NLCs.

## Figures and Tables

**Figure 1 pharmaceutics-14-01961-f001:**
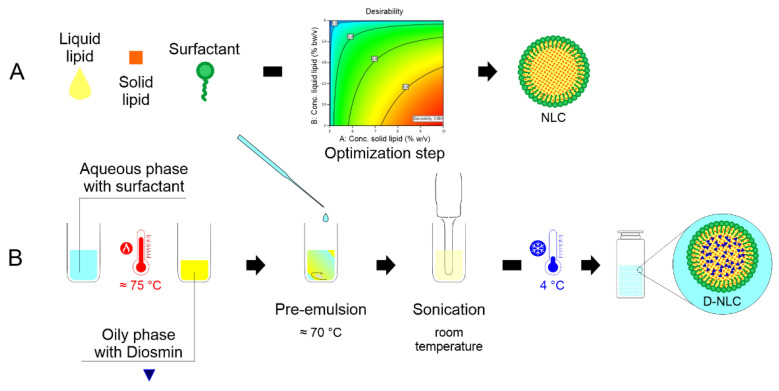
Scheme of production of blank NLCs (**A**) and diosmin-loaded nanoparticles (D-NLCs, (**B**)).

**Figure 2 pharmaceutics-14-01961-f002:**
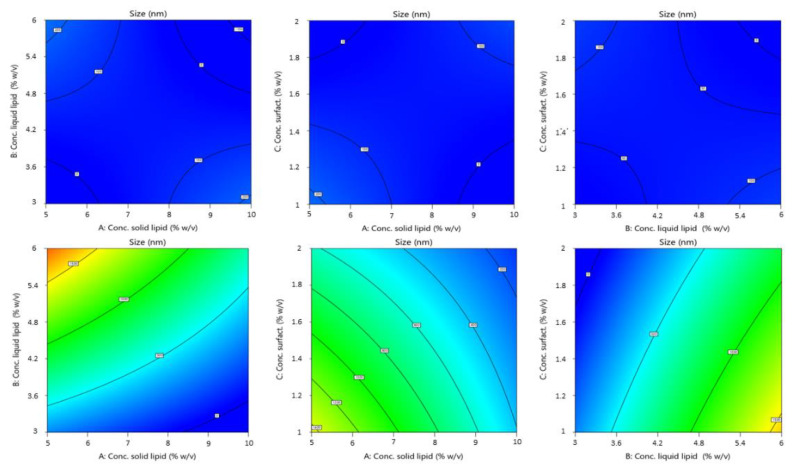
Contour plot of the effect of independent variables (from left to right: solid lipid concn. versus liquid lipid concn., solid lipid concn. vs. surfactant concn. and liquid lipid concn. vs. surfactant concn.) on the size of NLCs using Softisan^®^ 100 (top) and Gelucire^®^ 44/14 (bottom row).

**Figure 3 pharmaceutics-14-01961-f003:**
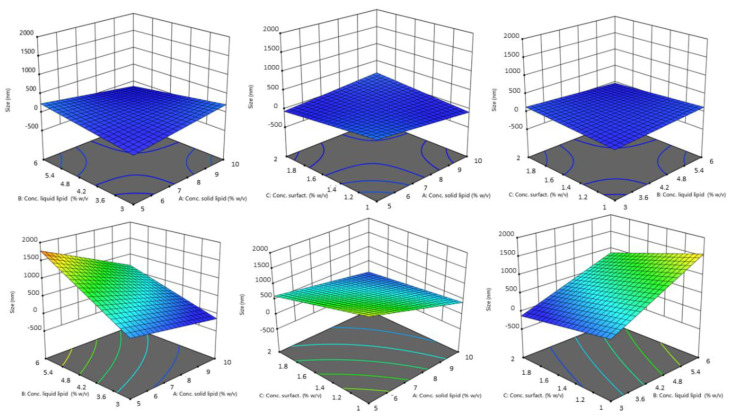
3D surface model of the effect of independent variables (from left to right: solid lipid concn. vs. liquid lipid concn., solid lipid concn. vs. surfactant concn., and liquid lipid concn. vs. surfactant concn.) on the size of NLCs using Softisan^®^ 100 (top) and Gelucire^®^ 44/14 (bottom row).

**Figure 4 pharmaceutics-14-01961-f004:**
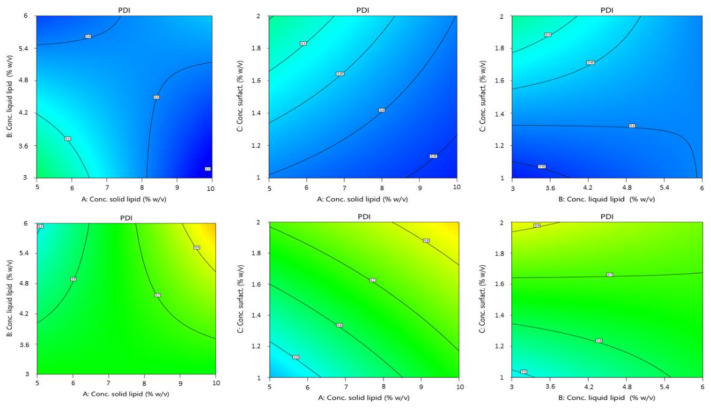
Contour plot of the effect of independent variables (from left to right: solid lipid concn. vs. liquid lipid concn., solid lipid concn. vs. surfactant concn., and liquid lipid concn. vs. surfactant concn.) on the PDI of NLCs using Softisan^®^ 100 (top) and Gelucire^®^ 44/14 (bottom row).

**Figure 5 pharmaceutics-14-01961-f005:**
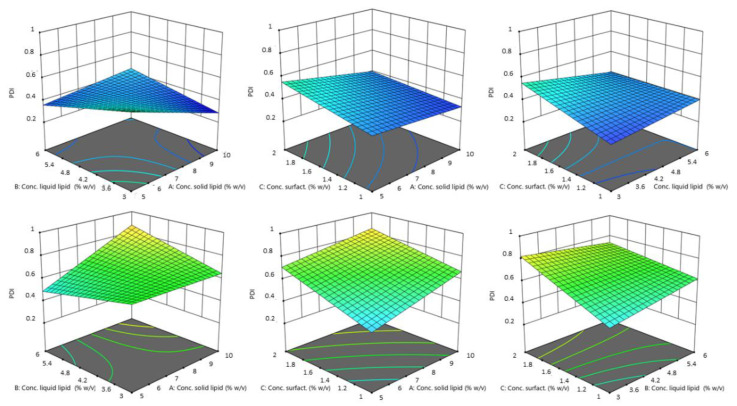
3D surface of the effect of independent variables (from left to right: solid lipid concn. vs. liquid lipid concn., solid lipid concn. vs. surfactant concn., and liquid lipid concn. vs. surfactant concn.) on the size of NLCs using Softisan^®^ 100 (top) and Gelucire^®^ 44/14 (bottom row).

**Figure 6 pharmaceutics-14-01961-f006:**
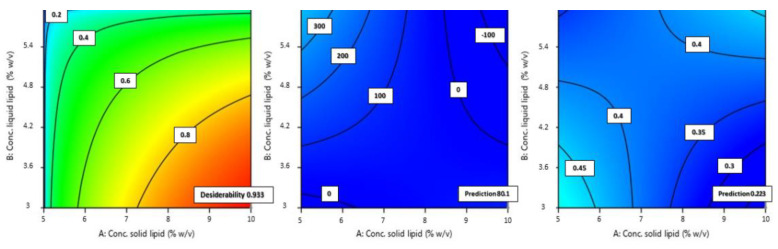
Contour plot of optimized formulation in terms of desirability, size, and PDI (from left to right).

**Figure 7 pharmaceutics-14-01961-f007:**
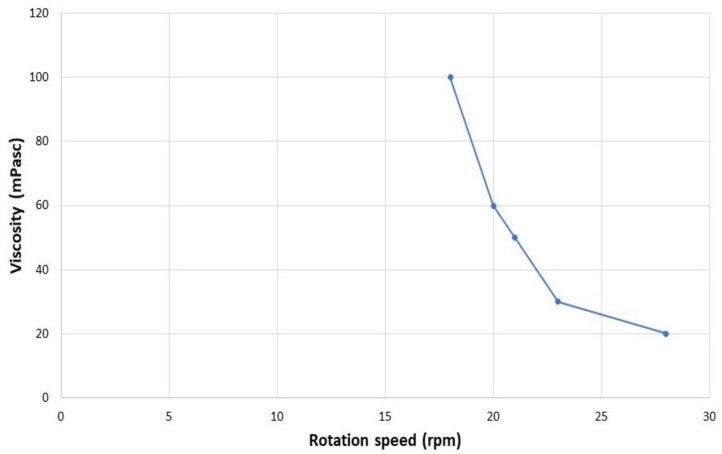
Effect of shear rate on viscosity (viscosity curve-pseudoplastic flow) of D-NLCs containing 0.15% (w/v) of HPMC.

**Figure 8 pharmaceutics-14-01961-f008:**
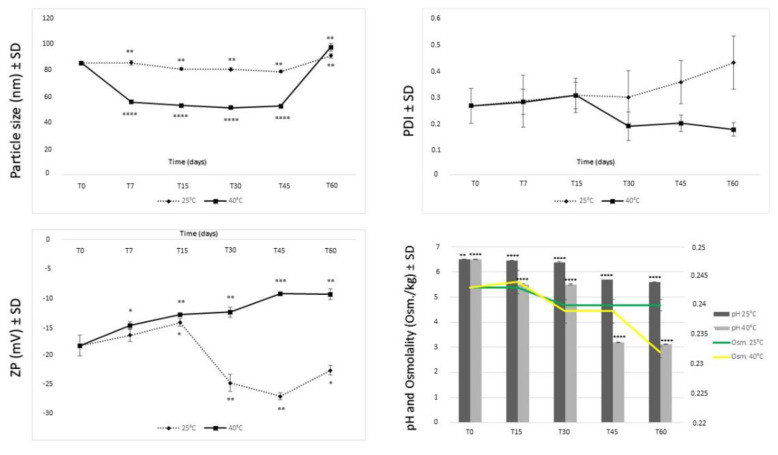
Stability in terms of mean particle size, PDI, zeta potential, pH, and osmolarity of NLCs at different storage conditions (25 °C/60% RH and 40 °C/75% RH). Significance was set as * *p* ≤ 0.05; ** *p* ≤ 0.01; *** *p* ≤ 0.001; **** *p* ≤ 0.0001.

**Figure 9 pharmaceutics-14-01961-f009:**
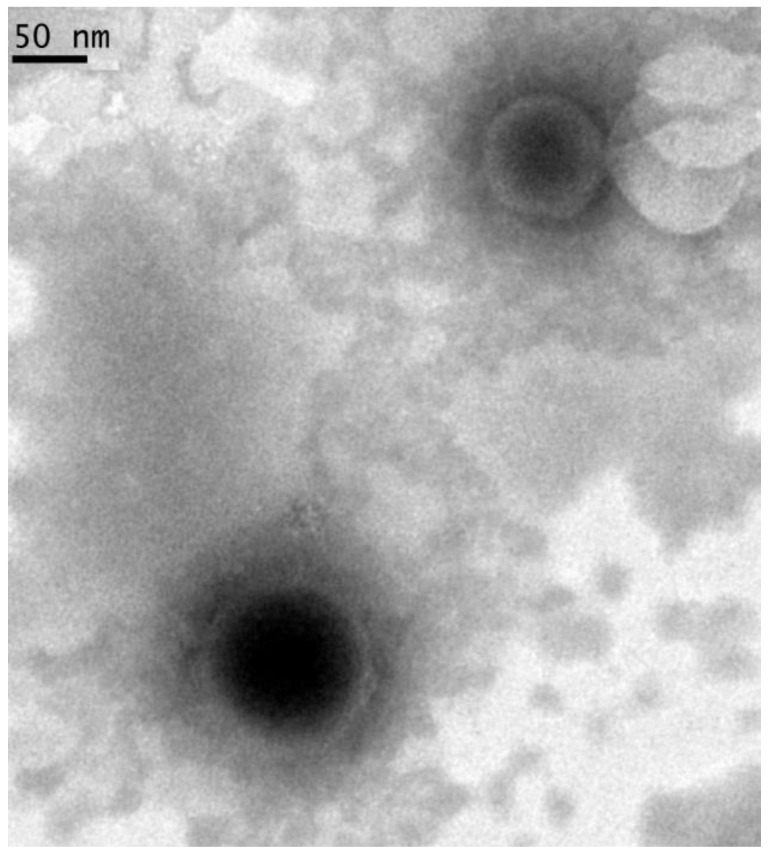
TEM image of D-NLCs.

**Figure 10 pharmaceutics-14-01961-f010:**
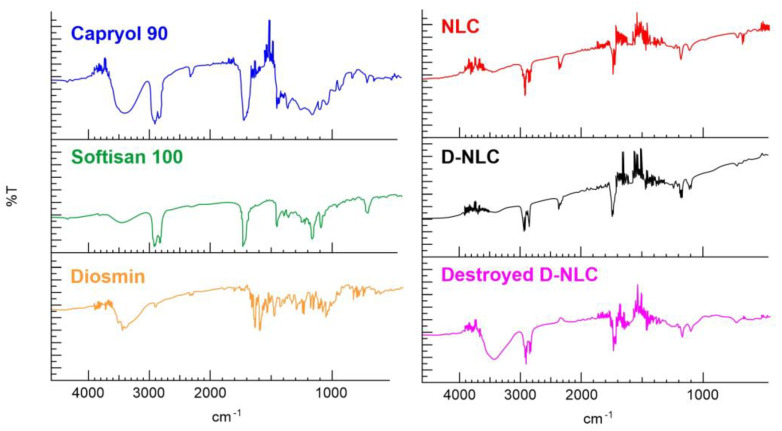
FT-IR analysis of the lipids used for the preparation of NLCs (Capryol^®^ 90 and Softisan^®^ 100), diosmin, blank NLCs, D-NLCs, and destroyed D-NLCs.

**Figure 11 pharmaceutics-14-01961-f011:**
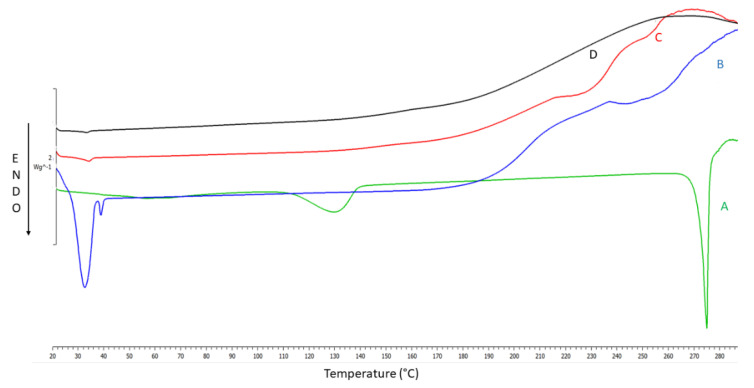
Differential scanning calorimetry (DSC) thermograms of diosmin (**A**), Softisan^®^ 100 (**B**), blank NLCs (**C**), and D-NLCs (**D**).

**Figure 12 pharmaceutics-14-01961-f012:**
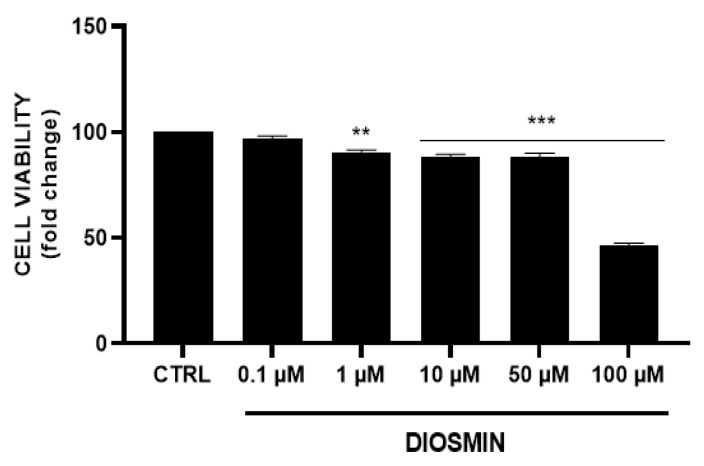
Assessment of diosmin effect on ARPE-19 cell viability (*** *p* < 0.0005, ** *p* < 0.005 vs. CTRL).

**Figure 13 pharmaceutics-14-01961-f013:**
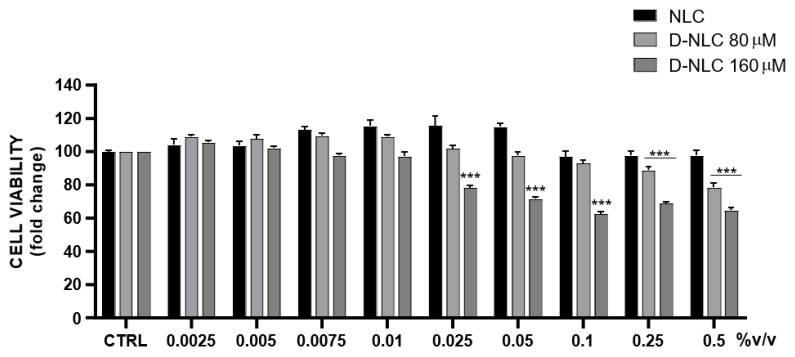
Evaluation of cytotoxicity of formulations of NLCs empty and loaded with different concentrations of diosmin (80–160 µM) on the ARPE-19 cell line (*** *p* < 0.0005 vs. NLCs).

**Figure 14 pharmaceutics-14-01961-f014:**
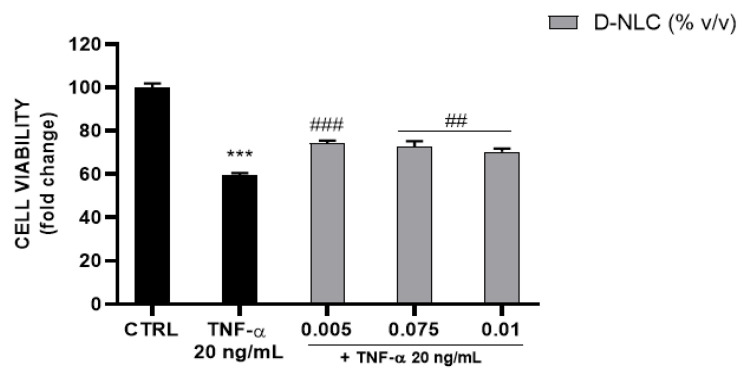
Evaluation of TNF-α (20 ng/mL) effect on ARPE-19 cell viability and recovery with co-treatment of TNF-α and D-NLC 160 µM (*** *p* < 0.0005 vs. CTRL; ### *p* < 0.0005, ## *p* < 0.005 vs. TNF-α).

**Table 1 pharmaceutics-14-01961-t001:** Independent variables of the experimental design.

Independent Variables	Units	Variable Type	Coded Factors	Levels	
				Low	High
**Solid lipid concn.**	(% w/v)	Numeric	X_1_	5	10
**Liquid lipid concn.**	(% w/v)	Numeric	X_2_	3	6
**Surfactant concn.**	(% w/v)	Numeric	X_3_	1	2
**Solid lipid type**		Categoric	X_4_	Gelucire^®^ 44/14; Softisan^®^ 100	

**Table 2 pharmaceutics-14-01961-t002:** Analysis of variance (ANOVA) for 2FI (two factors interaction) model. Term X1 corresponds to the solid lipid concentration, X2 to the liquid lipid concentration, X3 to the surfactant concentration, and X4 to the solid lipid type.

	Y_1_: Particle Size			Y_2_: PDI		
Source	F-Value	*p*-Value	Coefficient Estimate	F-Value	*p*-Value	Coefficient Estimate
**Model**	18.30	<0.0001	353.86	11.39	<0.0001	0.5433
X_1_ = solid lipid concn.	16.87	0.0009	−202.06	0.9433	0.3468	0.0194
X_2_ = liquid lipid concn.	32.81	<0.0001	281.75	0.0134	0.9093	−0.0023
X_3_ = surfactant concn.	8.36	0.0112	−142.21	17.92	0.0007	0.0844
X_4_ = solid lipid type	58.22	<0.0001	294.43	61.66	<0.0001	0.1229
X_1_X_2_	5.92	0.0279	−169.30	12.25	0.0032	0.0988
X_1_X_3_	3.64	0.0757	132.74	0.6361	0.4376	−0.0225
X_1_X_4_	13.36	0.0023	−179.78	13.81	0.0021	0.0741
X_2_X_3_	1.15	0.3001	−74.66	3.96	0.0652	−0.0561
X_2_X_4_	35.42	<0.0001	292.76	0.6686	0.4263	0.0163
X_3_X_4_	7.22	0.0169	−132.20	2.05	0.1727	0.0286

**Table 3 pharmaceutics-14-01961-t003:** NLC optimization process.

Factors	Goal	Limits
Solid lipid concn. (% w/v)	Maximize	5–10%
Lipid liquid concn. (% p/V)	Minimize	3–6%
Surfactant concn. (% w/v)	In range	1–2%
Type of solid lipid	Equal to Softisan^®^	Softisan^®^ 100, Gelucire^®^ 44/14
Size (nm)	Minimize	24.21–1960
PDI	Minimize	0.311–1.000

**Table 4 pharmaceutics-14-01961-t004:** Chemical and physical characterization of optimized NLCs.

Sample	Size (nm) ± SD	PDI ± SD	ZP (mV) ± SD	pH ± SD	Osmolality (Osm/kg) ± SD
OPT.NLC	85.33 ± 1.134	0.263 ± 0.067	−18.5 ± 1.8	6.51 ± 0.001	0.037 ± 0.00

**Table 5 pharmaceutics-14-01961-t005:** Technological characterization of D-NLCs at 80 µM and 160 µM.

Sample	Size (nm) ± SD	PDI ± SD	ZP (mV) ± SD	pH ± SD	Osmolality (Osm/kg) ± SD	EE% ± SD	Viscosity (mPa) ± SD
D-NLC (80 µM)	83.58 ± 0.77	0.278 ± 0.095	−18.5 ± 0.60	6.50 ± 0.010	0.253 ± 0.001	99.53 ± 2.50	23 ± 0.81
D-NLC (160 µM)	82.21 ± 1.12	0.267 ± 0.088	−18.0 ± 1.18	6.50 ± 0.005	0.251 ± 0.001	99.85 ± 2.05	23 ± 0.94

## Data Availability

Data generated during the study are available from the corresponding author.

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
