# Peer review of "Optimization of Lipid Nanoparticles by Response Surface Methodology to Improve the Ocular Delivery of Diosmin: Characterization and In-Vitro Anti-Inflammatory Assessment"

_pharmaceutics, 2022, doi:10.3390/pharmaceutics14091961_

Round 1
Reviewer 1 Report
Abstract:
Please indicate the outcome of the 60 day stability study
Introduction:
Line 47 – please specify, with references, which “main homeostatic mechanisms” are lost
Line 59-112 – this looks like one large paragraph. Please break it into ~4-6 smaller paragraphs.
Line 78 – “retinal inflammatory” looks like you are missing a word at the end of this.
Lines 113-114 – this does not belong in the introduction. If you would like to include it, please move it to your discussion or conclusion.
Materials and Methods:
Please have a clearer indication of which lipids are solid and which are liquid, and which is the surfactant (i.e. be clearer about the fact that Tween 80 is the surfactant). This could be differentiated in the materials section.
Section 2.4 – Please be clearer about what properties the “Desirability parameter” was looking out for. Was it size and PDI, or was the zeta potential considered too?
Lines 168-170 do not make sense – do you mean “subsequently” rather than “thus”? Saying 5 °C and thus room temperature indicates that 5 °C is room temperature.
Line 188 – it is unclear what the overall purpose of the viscosity measurement was and how exactly HPMC was added to adjust the viscosity. Was it dissolved and added dropwise? Was there a threshold viscosity you were trying to achieve? Was the same amount of HPMC added to each sample? Why was the rpm decreased rather than increased?
2.9 – the second paragraph again is very long and should be broken into several smaller paragraphs
Line 239 – why was Capryol 90 not analysed on its own? It is unclear why dissolution in dichloromethane was necessary.
2.11 – why did the stability study look at NLC and not D-NLC? It was assumed that drug loading and retention in the vesicles would be important to consider during storage.
Results and discussion
Lines 282-332 – another incredibly long paragraph, please break it down into smaller bits. There are several instances of this. Please ensure that all text throughout the document is arranged into bite-sized paragraphs that do not exceed 12 lines. Less than 10 lines would be ideal.
Lines 314-315 – “one of the most non-ionic surfactants used” does not make sense. Do you mean “one of the most used non-ionic surfactants”? Please correct this and the remainder of the sentence.
Lines 328-330 – the sentence here is generic by saying that the “results may vary”. It would be good to know how exactly LCT rather than MCT may affect the results.
Table 2 – please make the table heading more detailed to explain what all the variables mean.
3.1.1 and 3.1.2 headings – please correct the spelling of “independent”
Figures 1, 2, 3 and 4 – what is meant by “Conc. tens” on the y-axis? Also note spelling error durfactant instead of surfactant in the legend of Figures 1 and 2.
Line 411, 433, etc. – it would be easier to follow the text if the factors were spelt out rather than listed as A, B, C. The reviewer struggled to keep track of which factor corresponded to which letter.
Line 433 – “detained” does not appear to be the most appropriate word here.
Line 435 – what makes homogenisation “continuous” while probe sonication is not “continuous”? The authors indicated that the sonication they employed was in fact continuous.
Line 348, 471 (and elsewhere) – please don’t write “variance” or “variance analysis” after ANOVA. ANOVA already means “analysis of variance”.
Figure 5 – it is unclear based on the contour plots how the desirability of 0.993 fell at the same concentration as the size and PDI listed. Namely, the PDI appeared to be higher at that lipid concentration on the contour plot. It there a way to indicate where exactly on the contour plot the authors would like the reader to look? It is assumed that the same point will be indicated on each of the three plots.
Figure 6 – ideally viscosity should be on the y-axis and rpm on the x-axis.
Figure 7 – were any statistical evaluations performed?
Table 5 – could formulation viscosity also be shown here?
Figure 8 – should “destroyed-NLC” be “destroyed D-NLC”? The figure doesn’t clearly show that diosmin is present in this bottom right sample.
Figure 10 – why are the 80 and 160 µM D-NLC on different graphs rather than the same graph? The same NLC data is on both graphs.
Conclusion: A lot of introduction is reappearing here. It is suggested that the authors focus on their key findings and future directions.
Author Response
Reviewer 1
Comments and Suggestions for Authors
Abstract:
Please indicate the outcome of the 60 day stability study
Added, thank you
Introduction:
Line 47 – please specify, with references, which “main homeostatic mechanisms” are lost
The sentence was rewritten more clearly
Line 59-112 – this looks like one large paragraph. Please break it into ~4-6 smaller paragraphs.
Thank you, we went along the ms to revise the text accordingly
Line 78 – “retinal inflammatory” looks like you are missing a word at the end of this.
corrected, thanks
Lines 113-114 – this does not belong in the introduction. If you would like to include it, please move it to your discussion or conclusion.
The sentence was moved as suggested
Materials and Methods:
Please have a clearer indication of which lipids are solid and which are liquid, and which is the surfactant (i.e. be clearer about the fact that Tween 80 is the surfactant). This could be differentiated in the materials section.
The details required have been added to the text
Section 2.4 – Please be clearer about what properties the “Desirability parameter” was looking out for. Was it size and PDI, or was the zeta potential considered too?
Thank you for the suggestion, the desirability parameter was taken into account for size and PDI and not for Zeta Potential. Thus, it was specified in the section. (Lines 240-242)
Lines 168-170 do not make sense – do you mean “subsequently” rather than “thus”? Saying 5 °C and thus room temperature indicates that 5 °C is room temperature.
Corrected, thanks
Line 188 – it is unclear what the overall purpose of the viscosity measurement was and how exactly HPMC was added to adjust the viscosity. Was it dissolved and added dropwise? Was there a threshold viscosity you were trying to achieve? Was the same amount of HPMC added to each sample? Why was the rpm decreased rather than increased?
The purpose of the viscosity measurement is explained in section 3.4. The formulation already having a small size (~80 nm) could be subject to clearance, it is good that a viscosity higher than 0 improves retention on the ocular surface and avoids easy drainage.
The exact amount of HPMC added was also stated there. As suggested, it was also specified in section 2.8 how HPMC was added to the samples. The same amount of HPMC was added to each sample, trying the lowest amount first and then gradually adding more HPMC until the desired viscosity was reached.
2.9 – the second paragraph again is very long and should be broken into several smaller paragraphs
Done, thank you
Line 239 – why was Capryol 90 not analysed on its own? It is unclear why dissolution in dichloromethane was necessary.
Capryol 90 could also be analysed as it is. The use of dichloromethane was justified for improving the solubility of the oily ingredient and its spreadability onto the salt IR disk.
In addition, this solvent was chosen for its rapidity in evaporating unlike other solvents. The method (Thin Film Method using salt (NaCl disk) consisted of:
- Dissolve liquid sample in ~1-2 drops of dichloromethane
- Place 1 drop of this solution of one salt plate (only), let the solution evaporate, then proceed with FT-IR analysis as for a powder compound.
2.11 – why did the stability study look at NLC and not D-NLC? It was assumed that drug loading and retention in the vesicles would be important to consider during storage.
The stability study was only done for NLC and not D-NLC, as this work involves the optimization of a blank formulation and then loading the active compound into the optimized formulation. In fact, most of the chemical-physical and technological analyses, including the stability study, were done for the characterization of the optimized NLC formulation.
Stability at this time is also a preliminary analysis to evaluate the behaviour of the system during storage, at the moment without drug. In any case, subsequent work is planned in which all studies on the loaded formulation and further in-vitro analyses will be done, and this suggestion will be considered for the next manuscript. Thank you.
Results and discussion
Lines 282-332 – another incredibly long paragraph, please break it down into smaller bits. There are several instances of this. Please ensure that all text throughout the document is arranged into bite-sized paragraphs that do not exceed 12 lines. Less than 10 lines would be ideal.
Done, thank you
Lines 314-315 – “one of the most non-ionic surfactants used” does not make sense. Do you mean “one of the most used non-ionic surfactants”? Please correct this and the remainder of the sentence.
Corrected, thanks for the observation
Lines 328-330 – the sentence here is generic by saying that the “results may vary”. It would be good to know how exactly LCT rather than MCT may affect the results.
As suggested, a sentence has been added to specify how and what may vary in the two cases.
Table 2 – please make the table heading more detailed to explain what all the variables mean.
The heading was modified as requested
3.1.1 and 3.1.2 headings – please correct the spelling of “independent”
Done, thanks
Figures 1, 2, 3 and 4 – what is meant by “Conc. tens” on the y-axis? Also note spelling error durfactant instead of surfactant in the legend of Figures 1 and 2.
The words “conc. tens.” indicated the surfactant concentration. It was changed in all the relevant figures by inserting the terms “conc. surfact.” in the y-axis (indicating surfactant concentration) in accordance with the text. In addition, the word surfactant has been corrected in the figure captions.
Line 411, 433, etc. – it would be easier to follow the text if the factors were spelt out rather than listed as A, B, C. The reviewer struggled to keep track of which factor corresponded to which letter.
Thanks for the suggestion, the factors were renamed with the full name and/or factors X1-X4.
Line 433 – “detained” does not appear to be the most appropriate word here.
The term was modified
Line 435 – what makes homogenisation “continuous” while probe sonication is not “continuous”? The authors indicated that the sonication they employed was in fact continuous.
Thank you for your suggestion. By this sentence we meant that although it is a continuous process, because we chose to set “continuous” in the sonicator, it is conducted formulation by formulation, and in our case can lead to high variability when considered as a batch process. However, the sentence has been corrected and adjusted to give a clearer idea. Section 3.1.2.
Line 348, 471 (and elsewhere) – please don’t write “variance” or “variance analysis” after ANOVA. ANOVA already means “analysis of variance”.
Thank you, all the entries were revised
Figure 5 – it is unclear based on the contour plots how the desirability of 0.993 fell at the same concentration as the size and PDI listed. Namely, the PDI appeared to be higher at that lipid concentration on the contour plot. It there a way to indicate where exactly on the contour plot the authors would like the reader to look? It is assumed that the same point will be indicated on each of the three plots.
Thanking the reviewer for the comment, the desirability chosen (0.993) allows us to have a formulation that at the concentration of 10% (w/v) of solid lipid, 3% (w/v) of liquid lipid has a size value of 80.1 nm and a PDI of 0.223 nm. These values in the contour plot appear at the same point as the desirability because using these concentrations of lipid and surfactant, at this desirability value these two results should be obtained. Both desirability, size and PDI therefore can be found at the same point in the contour plot. The reader should look at the "prediction" (white window) on the contour plot. That is the point that indicates all three parameters (desirability, size and PDI), which although they are on three different graphs, are nevertheless at the same concentration of solid lipid and liquid lipid, indicating that at those concentrations we will obtain those values of desirability, size and PDI. Because of this desirability, we obtained values close to those desired, indicating a good correlation of the model.
Figure 6 – ideally viscosity should be on the y-axis and rpm on the x-axis.
As suggested, the graphic in figure 6 was modified reporting viscosity on the y-axis and rpm on the x-axis.
Figure 7 – were any statistical evaluations performed?
The standard deviation was added to the graphs. The level of significance has also been inserted in the graphs of stability studies, setting significance as ***p ≤ 0.001; ****p ≤ 0.0001. Thank you for the suggestion.
In paragraph 2.14 the method used for statistical analysis was also reported.
Table 5 – could formulation viscosity also be shown here?
Thank you for the suggestion, the viscosity values were added into the table.
Figure 8 – should “destroyed-NLC” be “destroyed D-NLC”? The figure doesn’t clearly show that diosmin is present in this bottom right sample.
Yes, there was an error, and it has been corrected in the Figure changing the title “destroyed-NLC” with the title “destroyed D-NLC” for the peak of the bottom right sample. Thank you.
Figure 10 – why are the 80 and 160 µM D-NLC on different graphs rather than the same graph? The same NLC data is on both graphs.
Thank you for your comment, the graph in actual figure 11 has been modified as you suggested.
Conclusion: A lot of introduction is reappearing here. It is suggested that the authors focus on their key findings and future directions.
The section was revised, thank you.
Reviewer 2 Report
The study, titled “Optimization of lipid nanoparticles by Response Surface Methodology to improve the ocular delivery of diosmin: characterization and in-vitro anti-inflammatory assessment” This manuscript discusses the optimization of D-NLC and characterized. The cytocompatibility of NLC with retinal epithelium was confirmed in vitro using ARPE-19 cells. I recommend the publication of this paper in Pharmaceutics after the major revision.
Comments:
1. The characterization of the optimized nanostructured lipid carriers NLC was limited. As a result, the added the following experimental results, such as XRD, TEM, DSC, and drug release studies, as suggested.
2. The manuscript should include a proposed illustration diagram of the D-NLC drug delivery system strategy. It would improve the manuscript's quality.
Author Response
Comments and Suggestions for Authors
The study, titled “Optimization of lipid nanoparticles by Response Surface Methodology to improve the ocular delivery of diosmin: characterization and in-vitro anti-inflammatory assessment” This manuscript discusses the optimization of D-NLC and characterized. The cytocompatibility of NLC with retinal epithelium was confirmed in vitro using ARPE-19 cells. I recommend the publication of this paper in Pharmaceutics after the major revision.
Comments:
- The characterization of the optimized nanostructured lipid carriers NLC was limited. As a result, the added the following experimental results, such as XRD, TEM, DSC, and drug release studies, as suggested.
Some additional analytical data have been added to better characterize the systems.
A DSC analysis has been added as you can see in the section 2.11 and 3.7
- The manuscript should include a proposed illustration diagram of the D-NLC drug delivery system strategy. It would improve the manuscript's quality.
As suggested, a graph has been added to schematize the NLC and D-NLC platform in section 2.5 (new Figure 1).